# Incrementally Adapting Generative Vision-Language Models with Task Codebook

## Abstract

With the help of large-scale pre-training, generative Vision-Language Models (VLMs) have acquired general-purpose capabilities. As downstream applications diversify, it is imperative for VLMs to learn and adapt continuously without experiencing catastrophic forgetting or necessitating complete retraining. In this work, we analyze the forgetting behavior of VLMs and propose a solution to enhance their incremental learning abilities. We introduce a Task Codebook within VLMs, enabling efficient retrieval of task-specific parameters for model adaptation. Our evaluation encompasses a diverse set of tasks spanning a wide range of visual domains and textual instructions. Experiments demonstrate that our approach effectively mitigates forgetting, even under highly demanding task sequences.

## 1 Introduction

Recent advancements in generative vision-language models OpenAI (2023); Alayrac et al. (2022); Chen et al. (2023c); Wang et al. (2022a); Google (2024) (VLMs) have demonstrated remarkable success. These models leverage extensive pre-training corpora to acquire substantial world knowledge, facilitating adaptation to downstream tasks. This paper investigates the *incremental learning* capabilities of such models. Our objective is to incrementally adapt generative VLMs to multiple tasks, rather than maintaining separate models specialized for each task.

A significant challenge in incremental learning is *catastrophic forgetting* McCloskey & Cohen (1989), where models rapidly lose previously learned knowledge. An ideal incrementally trained model should mitigate forgetting while retaining the ability to acquire new knowledge. Existing research has primarily focused on *class-incremental learning* for classification, assuming sequential availability of class subsets.

While class-incremental learning is an important research direction, general-purpose VLMs span a much wider set of applications than simple classification. It is thus more realistic to consider a *diverse* set of applications ranging from classification over detection to question answering. This poses a greater challenge for the incremental learning setting, as the model now needs to learn new applications instead of expanding a single application's coverage. More recent studies have examined catastrophic forgetting in vision-language models Garg et al. (2023); Zheng et al. (2023); Zhang et al. (2023). However, these methods often assume task similarity (e.g., different VQA tasks) He et al. (2023), limiting their practical applicability.

We propose a new incremental learning method for generative VLMs that improves their ability to learn and adapt to various, and potentially very different, tasks over time. To that end, we introduce a *task codebook* that stores *adapters* specialized for each encountered task. When presented with a new task, the model learns to access its most relevant modules to enhance its performance. This design allows our model to effectively learn diverse tasks without needing to know at inference time which task it is being asked to do, all while avoiding the common issue of *catastrophic forgetting* Compared to existing prompt-based solutions Wang et al. (2022c), we find that our method is more flexible and performs well across a wide set of incremental-learning settings.

To comprehensively evaluate the incremental learning capabilities of generative VLMs, we present a novel benchmark comprising diverse tasks, such as captioning Young et al. (2014); Gurari et al. (2020), VQA Goyal et al. (2017); Marino et al. (2019), OCR-enhanced captioning and VQA Sidorov et al. (2020); Wang et al. (2021), open-vocabulary classification Wang et al. (2022a), object detection Lin et al. (2014), referring expression generation Kazemzadeh et al. (2014), grounding Kazemzadeh

et al. (2014) and multi-modal instruction tuning Liu et al. (2023). This diverse set of tasks enables a thorough assessment of incremental learning abilities under varying conditions, including different classes, datasets, and applications.

Our contributions in this paper can be summarized as follows:

- We propose a novel adaptation strategy for generative models with a task codebook that allows lookup and dynamic routing at inference time. This approach demonstrates efficacy in scenarios involving sequential tasks of different natures.
- We propose a new multi-modal incremental learning benchmark for generative models, spanning 36 datasets and 8 applications.
- We provide a comprehensive evaluation of our method under various scenarios, demonstrating our method's superiority over existing baselines across a wide range of tasks.

## 2 RELATED WORK

**Incremental learning.** In this task, the assumption is that the training set of a dataset is not available all at once. Data from previous timestamps are discarded as new data becomes available. Existing studies mainly focus on single modality such as images Hsu et al. (2018); Van de Ven & Tolias (2019) or texts Ke et al. (2023). For incremental learning in the vision domain, typical incremental learning setups include: task-incremental learning Hsu et al. (2018); Van de Ven & Tolias (2019), class-incremental learning Hsu et al. (2018); Van de Ven & Tolias (2019), domain-incremental learning Hsu et al. (2018); Van de Ven & Tolias (2019), task-agnostic incremental learning Aljundi et al. (2019a), and online continual learning Aljundi et al. (2019b).

**Class-incremental learning.** Existing approaches address this challenge through regularization techniques such as knowledge distillation Li & Hoiem (2017), model expansion Rusu et al. (2016); Wang et al. (2017) or representations Yan et al. (2021) and weight consolidation Aljundi et al. (2018); Chaudhry et al. (2018); Kirkpatrick et al. (2017). Alternatively, replay-based methods retain a small sample of data from previous classes Rebuffi et al. (2017); Chaudhry et al. (2019); Iscen et al. (2020). More recent techniques introduce prompt retrieval for class-incremental learning Wang et al. (2022c;b); Khan et al. (2023); Tang et al. (2023) without requiring a replay buffer.

**Multi-modal incremental learning.** Multi-modal incremental learning has seen recent interest due to the advancement of multi-modal models Radford et al. (2021); Wang et al. (2022a); Li et al. (2023). Some of the research has focused on dual-encoder models, *e.g.* CLIP Radford et al. (2021), examining their performance on evolving data distributions Garg et al. (2023) and knowledge preservation Zheng et al. (2023). Other studies focus on incremental learning for the Visual-Question Answering (VQA) tasks, either by formulating it as a classification problem Qian et al. (2023); Srinivasan et al. (2022) or a generative task Zhang et al. (2023); He et al. (2023). In this paper, we adopt a more practical setting with varied tasks and leverage generative models like GIT Wang et al. (2022a) for their potential across diverse settings.

**Codebook learning.** Codebook learning is an effective method for converting continuous features into semantically rich, discrete codes. This technique has found widespread application in the field of self-supervised representation learning Caron et al. (2020; 2021) and reconstruction Van Den Oord et al. (2017); Ramesh et al. (2021). Inspired by recent advancements in retrieval mechanisms that facilitate incremental learning in classifiers Prabhu et al. (2023); Wang et al. (2022c;b), this study introduces the concept of a task codebook. Our approach enables sequential task learning in a non-interfering manner, effectively addressing the challenge of catastrophic forgetting.

## 3 METHOD

Our goal is to incrementally adapt a generic auto-regressive model to various challenging (and potentially very diverse) image-to-text tasks. In this section, we start by defining the task incremental learning setup before describing our proposed task codebook incremental adaptation (TCIA) method.

### 3.1 PROBLEM SETUP

Auto-regressive vision-language models form the building blocks of most state-of-the-art systems for challenging image-to-text tasks Chen et al. (2023b). While a large body of work has focused on efficient ways to adapt these models to different tasks Hu et al. (2021); Lester et al. (2021); Li & Liang (2021), we explore in this work how to adapt them to a sequential stream of tasks, in an

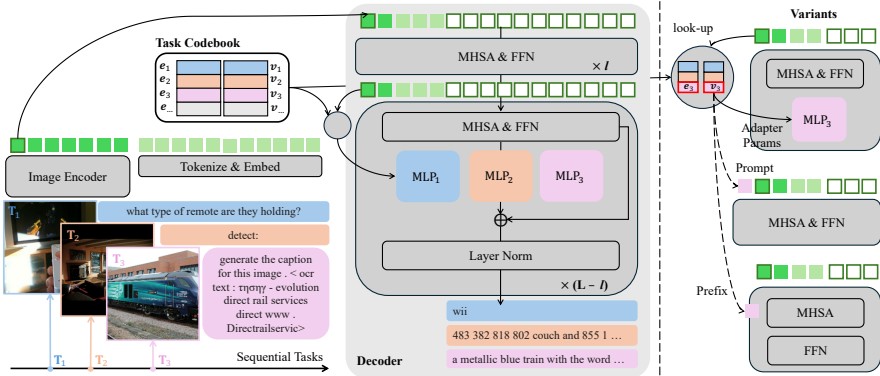

Figure 1: **Overview of TCIA**. We introduce a task codebook in a generic auto-regressive generative model. Task-specific codebook keys are used to retrieve task-specific codebook values, which are adapter MLP parameters Houlsby et al. (2019) by default (**left**). We also consider prompt Lester et al. (2021); Wang et al. (2022c) or prefix Li & Liang (2021) as alternative ways to condition the model into different tasks (**right**).

*incremental* manner. Formally, let us consider a task mixture $\mathbf{T}$ of $T$ different tasks: a data sample for each task $t \in \mathbf{T}$ is represented as $x \sim \mathcal{D}_t$, such that $x = (x_v, x_i, y) \sim (\mathcal{V}_t, \mathcal{I}_t, \mathcal{Y}_t)$, where $\mathcal{V}_t, \mathcal{I}_t$ and $\mathcal{Y}_t$ respectively denote the visual input, textual instruction input and textual target output spaces of this task. For example, when considering a VQA task, $\mathcal{V}_t$ represents images, $\mathcal{I}_t$ questions and $\mathcal{Y}_t$ expected answers for this task.

Typically, models are adapted to each task $t$ individually, hence resulting in a collection of $T$ specialized models. We refer to this strategy as *single-tasking*. On the other hand, it is also common to train a single model on the entire task mixture $\mathbf{T}$ at once by, for example, randomly sampling a different task from the mixture each step. We refer this as *multi-tasking*. In this work, we consider another scenario where tasks come in a sequential manner. This mirrors real-world conditions where practitioners adapt a single model on the fly due to the unavailability of all tasks upfront.

**Sequential tasks.** We consider the scenario where tasks arrive sequentially, requiring continuous adaptation of a single model. Specifically, for task $t$, we assume access solely to $\mathcal{D}_t$ (data from the corresponding task), and not to $\mathcal{D}_i$ for any $i \neq t$ (data from the other tasks). The challenge of this *incremental learning* setup is to learn over the different tasks while mitigating *catastrophic forgetting*. In contrast to previous incremental learning methods focusing on incrementally growing label spaces for image recognition Wang et al. (2022b;c), we assume that the nature of the task and the data distribution $\mathcal{D}_t$ can vary significantly across tasks.

**Auto-regressive generative models.** Since our goal is to solve a mixture of image-to-text tasks, we build upon standard auto-regressive generative vision-language models Wang et al. (2022a). Formally, let us consider an input image-text data pair $x = (x_v, x_i) \sim (\mathcal{V}_t, \mathcal{I}_t)$ from task $t$. It is transformed into a set of $N$ $d$-dimensional embeddings $\mathbf{X} \in \mathbb{R}^{N \times d}$ formed by concatenating the output of a visual encoder processing $x_v$ and a text tokenizer processing $x_i$. We use an auto-regressive text decoder $g(\cdot)$ with $L$ layers to generate the target text $y \in \mathcal{Y}_t$. Specifically, the decoder predicts each text token $y_k$ from the target text (we denote its length by $K$) given both the set of preceding token embeddings $\mathbf{Y}_{<k}$ and the input image-text pair embeddings $\mathbf{X}$. We train with a language modeling objective:

$$\mathcal{L} = \frac{1}{K} \sum_{i=1}^{K} \ell(y_k, g([\mathbf{X}; \mathbf{Y}_{y_{<k}}]))$$

where $[;]$ corresponds to the concatenation operation in the first dimension and $\ell$ is the softmax cross-entropy loss with label-smoothing Müller et al. (2019). We average this loss over minibatches of examples drawn in $(x_v, x_i, y) \sim (\mathcal{V}_t, \mathcal{I}_t, \mathcal{Y}_t)$.

## 3.2 TASK CODEBOOK INCREMENTAL ADAPTATION (TCIA)

Incremental learning often employs buffer replay, using either past input data Chaudhry et al. (2019) or past model weights Li & Hoiem (2017) to mitigate catastrophic forgetting. However, storing extensive data or numerous model replicas becomes infeasible as task and model scales increase. In

this paper, we introduce a novel task codebook method using task-specific keys and values. The task keys are efficiently retrieved using a nearest neighbor lookup mechanism and dynamically updated. Each retrieved task key is paired with a task value, which is then employed to efficiently condition the remaining layers of the model Lester et al. (2021); Li & Liang (2021); Houlsby et al. (2019) to the considered task. This design choice effectively integrates our task lookup module within the decoder, enabling the deeper generative layers to adapt to a diverse set of tasks. An overview of our method is presented in Fig. 1.

**Task codebook.** Formally, we introduce a task codebook at the $l^{\text{th}}$ decoder layer of the generative model. This codebook contains a key-value pair $(e_t, v_t)$ for each task $t$ of the task mixture $\mathbf{T}$. The task-specific keys $e_t \in \mathbb{R}^d$ are used to recognize which task to be dealt with while the task-specific values $v_t$ are employed to adapt the last $N_l = L - l$ layers of the model to the considered task.

**Task-specific keys: recognizing tasks with lookups.** Intuitively, we want a task-specific key $e_t$ to be able to recognize which task $t$ we are currently solving, making it play the role of a *prototype* for task $t$. A simple yet effective strategy to do so is to use an average of the input embeddings coming from the considered task $t$. Specifically, at training time, we dynamically update the relevant task keys with an exponential moving average (EMA) of the sequence representation at the $l^{\text{th}}$ decoder layer, denoted by $\mathbf{x}^l$, coming from examples sampled from $\mathcal{D}_t$. Formally, we have for each task $t$: $e_t = m \cdot e_t + (1 - m) \cdot \mathbf{x}^l$ where $\mathbf{x}^l$ is the embedding of an example $x = (x_v, x_i) \sim (\mathcal{V}_t, \mathcal{I}_t)$ sampled from task $t$ and $m$ is a decay rate. An alternative is to learn the task keys via gradient descent using an additional loss term as in L2P Wang et al. (2022c). Our simple EMA design offers faster adaptation and results in improved performance compared to learning the keys (see Sec. 5.5). We empirically find that max-pooling exhibits robust advantages across all learning setups when it comes to the choice of the sequence representation (see also Sec. 5.5).

At inference time, we simply select a relevant task index $t$ for the evaluated input $x$ by performing a nearest-neighbor lookup, defined as $t = \text{argmax}_i \gamma(e_i, \mathbf{X}_0^l)$ where $\gamma$ is a similarity metric. In practice, we use cosine similarity.

**Task-specific values: parameter-efficient adaptation.** In order to achieve efficient adaptation to each task and minimal parameter overhead, we integrate the task-specific values $v_t$ as parameters within an *adapter* module Houlsby et al. (2019). Specifically, the task-specific values $v_t$ represent a set of $N_l$ lightweight bottleneck multi-layer perceptron (MLP) parameters introduced after the attention and feedforward layers of each of the last $N_l$ transformer decoder blocks. We refer to this way of adapting the decoder using task-specific MLPs by the *adapter* strategy and denote the resulting model by TCIA$^{\text{A}}$. As illustrated in the right panel of Fig. 1, we also evaluate alternative strategies denoted by TCIA$^{\text{P}}$ (resp. TCIA$^{\text{Pr}}$) where task-specific values $v_t$ refer instead to a prompt Lester et al. (2021); Wang et al. (2022c) (resp. to a prefix Li & Liang (2021)). We find in our experiments (see Tab. 1) that while these two variants perform competitively and strongly allieviate forgetting, our variant using adapters, *i.e.* TCIA$^{\text{A}}$, leads to the best version of our task codebook incremental adaptation framework.

### 3.3 IMPLEMENTATION DETAILS

**Auto-regressive model.** We build upon GiT-Large Wang et al. (2022a), a decoder-only auto-regressive model with 400M parameters. The image encoder is initialized from CLIP-L/14 Radford et al. (2021) and we use a 6-layer text decoder with internal dimension $d = 768$. Our model is first pre-trained jointly on WebLI-100M Chen et al. (2023c) and Conceptual Captions-12M Sharma et al. (2018) as a general image captioner.

**Task codebook.** We insert the task codebook before the first layer of the decoder, *i.e.* we have $l = 0$. For adapters Houlsby et al. (2019), we use two 2-layer MLP modules with an internal dimension $s = 256$, resulting in a parameter increase of $N^l \times T \times 2(2ds + s + d)$. The parameter count increase for the other variants of TCIA are $N^l \times T \times ds$ for TCIA$^{\text{P}}$ and $N^l \times T \times 2ds$ for TCIA$^{\text{Pr}}$. The decay rate $m$ is set to 0.99 by default.

**Upper-bounds.** Our upper-bounds are *single-tasking* and *multi-tasking*. These variants do not adapt the model sequentially, but either train multiple specialized models (single-tasking) or a general model with all tasks at once (multi-tasking).

**Baselines.** For sequential task learning, we consider a simple baseline which consists in simply fine-tuning the model to different tasks in a sequential manner. This usually leads to catastrophic forgetting. Episodic Replay (E-Replay) Chaudhry et al. (2019) is another baseline where we store

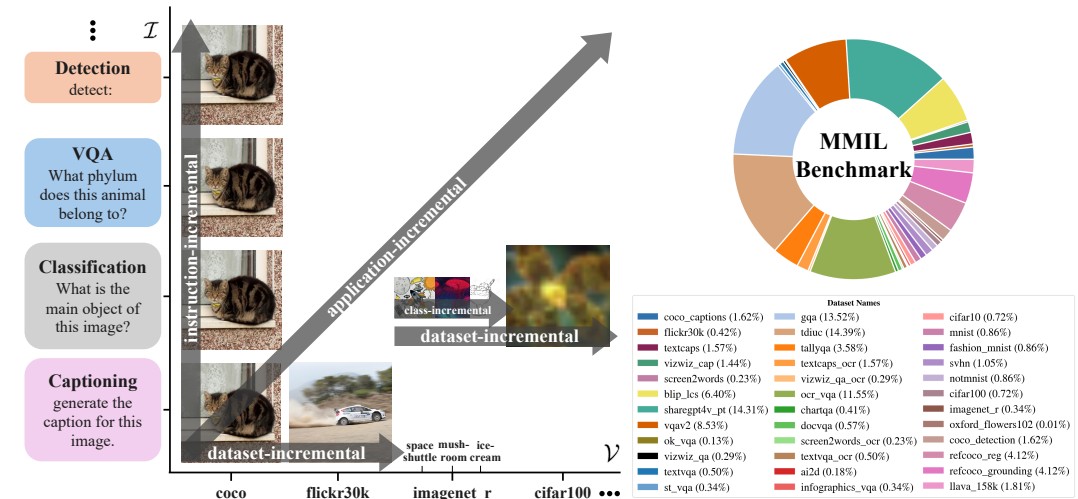

Figure 2: **Multi-Modal Incremental Learning (MMIL) benchmark** consists of 36 datasets across 8 applications (**left**) and 4 incremental learning setups (**right**).

previous samples in a memory buffer to re-use when learning new tasks. This constitutes a strong and effective baseline for sequential tasks. We use 1% of the entire dataset as the size of the memory buffer by default, similar to He et al. (2023). Finally, we extend L2P Wang et al. (2022c), a state-of-the-art incremental learning method designed for classification, to our multi-modal benchmark , by using the concatenated visual feature and textual instruction feature from CLIP-B Radford et al. (2021) as the *key*. We call it L2P+. Prompts are then retrieved from the pool and used as additional input to the decoder.

## 4  MULTI-MODAL INCREMENTAL LEARNING BENCHMARK

In this section, we first introduce the comprehensive set of tasks employed in our benchmark. We then define different incremental learning paradigms utilized in our experiments. A schematic diagram is demonstrated in Fig. 2.

### 4.1  APPLICATIONS

To assess the adaptability of VLMs in addressing a wide set of applications, we curate a benchmark of 8 multi-modal applications, including *captioning*, *VQA*, *OCR-CV*, *classification*, *detection*, *referring expression*, *grounding*, and *multi-modal instruction tuning*, across a total of 36 datasets. These applications span a wide range of visual and textual domains detailed in  Appendix A.1 and Tab. 7.

### 4.2  INCREMENTAL-LEARNING SETUPS

In Sec. 3, we describe our method which adapts the model for each task $t$ in a task mixture $\mathbf{T}$. We now define various incremental-learning setups with differing definitions for the mixture $\mathbf{T}$ and showcase their connections in Fig. 2.

**Dataset-incremental learning.**  In this setup, we define each task $t$ as a single dataset. The task mixture $\mathbf{T}$ refers to the union of different datasets within the same application. For example, classification is an application and CIFAR-100 and ImageNet-R datasets are different tasks for this application.

**Class-incremental learning.**  This is the traditional incremental classification setup Wang et al. (2022c;b); Khan et al. (2023), where each task $t$ refers to a subset of classes of a classification dataset. The entire task mixture $\mathbf{T}$ corresponds to a single dataset, such as CIFAR-100 in this scenario.

**Instruction-incremental learning.**  We fix the visual input set $\mathcal{V}$ in this scenario and incrementally modify the textual input $\mathcal{I}$ over different tasks. We use the COCO dataset Lin et al. (2014) and adapt the model with different textual instructions in each task $t$. Different variations of textual instructions include captioning, VQA (OK-VQA Marino et al. (2019)), object detection, referring expressions (RefCOCO Kazemzadeh et al. (2014)) and grounding (RefCOCO Kazemzadeh et al. (2014)). The respective textual inputs for these five tasks are the following: *"Generate the caption for this image."*,

Table 1: **Dataset-incremental learning** where different tasks correspond to different datasets of the same application. The considered applications are captioning, VQA, OCR-CV, and classification. *F* and *Acc.* denote forgetting and average accuracy at the end of training. All numbers are run by us.

| Method | Captioning | | VQA | | OCR-CV | | Classification | |
| --- | --- | --- | --- | --- | --- | --- | --- | --- |
| | F↓ | Score ↑ | F↓ | Score ↑ | F↓ | Score ↑ | F↓ | Score ↑ |
| *Single-tasking* | | | | | | | | |
| Finetuning | – | 121.4 | – | 56.7 | – | 49.1 | – | 97.1 |
| Adapter | – | 104.7 | – | 56.4 | – | 48.4 | – | 94.7 |
| *Multi-tasking* | | | | | | | | |
| Finetuning | – | 112.9 | – | 59.4 | – | 40.4 | – | 96.9 |
| Adapter | – | 96.9 | – | 56.8 | – | 37.4 | – | 93.8 |
| *Sequential tasks* | | | | | | | | |
| Finetuning | 26.4 | 62.3 | 7.97 | 42.4 | 20.4 | 14.8 | 71.3 | 24.2 |
| Adapter | 12.1 | 57.5 | 13.6 | 36.4 | 20.1 | 9.63 | 82.0 | 11.9 |
| E-Replay | 15.9 | 77.3 | 3.67 | 52.7 | 4.25 | 30.3 | 23.0 | 67.0 |
| L2P+ | 3.14 | 83.2 | 0.55 | 48.9 | 1.50 | 35.4 | 0.39 | 90.8 |
| TCIA$^P$ (Ours) | 0.66 | 79.4 | 0.66 | 50.7 | 0.38 | 35.3 | 0.08 | 90.9 |
| TCIA$^{Pr}$ (Ours) | 0.84 | 88.1 | 0.93 | 53.2 | **0.23** | 39.2 | **0.06** | 92.5 |
| TCIA$^A$ (Ours) | **0.47** | **100.8** | **0.54** | **55.1** | 0.31 | **45.7** | 0.15 | **94.0** |

*"<question>"*, *"detect:"*, *"describe the box at <location>"* and *"detect <object>:"*. Overall, the task mixture $\mathbf{T}$ is the COCO dataset seen five times, each associated with textual inputs of a different nature.

**Application-incremental learning.** This setup increments over different *applications*. For each task $t$, we adapt the model to a specific application, *e.g.* VQA, by training on the combined datasets corresponding to that application. The union of all tasks *i.e.* $\mathbf{T}$ refers to the combination of 8 applications (see Section 4.1) in this scenario.

## 4.3 METRICS

To evaluate the efficacy of TCIA and the baseline methods, we employ distinct metrics appropriate to each dataset and application. We report the average of all the metrics across different datasets and tasks as correctness *score* (higher is better). We also report the widely-used *forgetting* metric Chaudhry et al. (2018) (denoted by F in our experiments), which quantifies the extent to which a model has forgotten a task based on its current state. See Appendix A.4 for detailed definition.

## 5 EXPERIMENTS

In this section, we first assess the performance of our proposed TCIA framework on the different incremental learning setups described in Sec. 4.2. Second, we show an ablation study of different important components of TCIA. Finally, we identify potential areas of improvement for our model by analyzing the performance of an oracle model that always chooses the right task at inference.

## 5.1 DATASET-INCREMENTAL LEARNING

We conduct dataset-incremental learning across 4 diverse applications. These applications are all comprised of different datasets as detailed in Tab. 7.

**Baseline comparison.** We see in Tab. 1 that TCIA models outperform all the considered baselines in the sequential task learning scenario. First, as expected, the naive baseline of training the model directly on sequential tasks yields severe forgetting: for example we observe $-59.1$ points of performance drop on captioning compared to single-tasking for finetuning. Second, we see that the strong E-Replay Chaudhry et al. (2019) and L2P+ Wang et al. (2022c) baselines alleviate partly the forgetting but still suffer from performance degeneration. The two main design differences between TCIA$^P$ and L2P+ methods are (i) exponential moving average (EMA) versus gradient back-propagation to update the task-specific keys and (ii) the task-specific keys are built from decoder inner layers rather than off-the-shelf backbones Dosovitskiy et al. (2020). We validate in Tab. 6i that our design choices are favorable. Details of incremental learning scores across training steps are provided in the Fig. 6. Overall, our TCIA models improve over strong baselines for incremental learning, especially on complex multi-modal benchmarks.

Table 2: **State-of-the-art comparison in class-incremental learning.** 10 class splits of CI-FAR100 Krizhevsky et al. (2009); Wang et al. (2022c) and Split ImageNet-R Wang et al. (2022b) are considered. *Add. Models* denotes the additional models used. For E-Replay Chaudhry et al. (2019), we specify the size of the memory buffer in parenthesis. For reference, we also report the upper bound by training on the entire dataset, corresponding to *multi-tasking*. We bold best results within the two studied model size categories: *base* and *large*.

| | Method | Model | Add. Models | Split CIFAR100 | | Split ImageNet-R | |
|---|---|---|---|---|---|---|---|
| | | | | F↓ | Score ↑ | F↓ | Score ↑ |
| Base | Upper-bound | ViT-B | – | – | 90.9 | – | 79.1 |
| | Upper-bound | GiT-B | – | – | 85.9 | – | 80.0 |
| | E-Replay (1k) | ViT-B | – | 33.3 | 67.9 | 35.4 | 55.1 |
| | E-Replay (5k) | ViT-B | – | 16.5 | 82.5 | 23.3 | 65.2 |
| | L2P | ViT-B | ViT-B Dosovitskiy et al. (2020) | 7.35 | 83.9 | 9.73 | 61.6 |
| | DualPrompt | ViT-B | ViT-B Dosovitskiy et al. (2020) | 5.16 | 86.5 | 4.68 | 68.1 |
| | LGCL | ViT-B | VIT-B Dosovitskiy et al. (2020) & CLIP-B Radford et al. (2021) | **5.10** | **87.2** | 4.20 | 69.5 |
| | TCIA | GiT-B | ViT-B Dosovitskiy et al. (2020) | 9.25 | 84.6 | **3.70** | **72.9** |
| Large | Upper-bound | GiT-L | – | – | 94.1 | – | 93.5 |
| | E-Replay (5k) | GiT-L | – | 12.9 | 84.9 | 12.5 | 71.5 |
| | TCIA | GiT-L | – | 10.2 | 81.7 | 12.1 | 59.5 |
| | TCIA | GiT-L | ViT-B Dosovitskiy et al. (2020) | **9.16** | **87.0** | **3.02** | **77.6** |

**Different TCIA variants.** We observe in the last three rows of Tab. 1 that our model with adapters (TCIA$^A$) leads to better performance across different dataset-incremental learning benchmarks. The superiority of the adapter variant compared to prompt or prefix variants is mainly explained by the stronger transfer capability of the adapter module in general. Interestingly, we observe that TCIA$^P$ and TCIA$^{Pr}$ models also perform strongly and exhibit low forgetting. This shows that our TCIA framework is robust to the choice of adaptation method. In the following experiments, we stick to the best performing TCIA$^A$ variant and denote it as TCIA for brevity.

## 5.2 CLASS-INCREMENTAL LEARNING

Further, We apply our TCIA framework to the standard class-incremental learning scenario. This allows us to compare our results with previously published numbers. Note that unlike existing methods Wang et al. (2022c); Khan et al. (2023), TCIA is not designed specifically for discriminative classification tasks. Yet, we show that our generative TCIA model is still effective and performs decently well. To tailor our method for classification, we further consider using features from an additional model (*Add. Model*) to learn task key representation. We further consider varying class splits in the Appendix B.2, showcasing the unique advantage of TCIA when class splits increase.

**Comparison with the state of the art.** We compare TCIA to previously published results in the traditional setting of class-incremental learning on CIFAR-100 and Split ImageNet-R Wang et al. (2022c). In Tab. 2, we see that TCIA reaches competitive performance with recent state-of-the-art class-incremental learning, though it is not specifically designed for this scenario. Notably, we improve over previously published work Khan et al. (2023) on Split ImageNet-R by +8.1 points (77.6 *v.s.* 69.5). Our performance on CIFAR100 is competitive but remains -0.2 points behind the state of the art (87.0 *v.s.* 87.2), likely because CIFAR100 is a small dataset with limited visual and task complexity and hence does not benefit from the potential of adapting large multi-modal models. TCIA exhibit less forgetting compared to the E-Replay baseline as an increasing amount of classes are presented, especially for a more complicated dataset like Split ImageNet-R.

## 5.3 INSTRUCTION-INCREMENTAL LEARNING

In this experiment, we consider the instruction-incremental learning setup as detailed under the corresponding paragraph in Sec. 4.2. The goal is to incrementally learn to perform different tasks, namely captioning, VQA, object detection, referring expressions and grounding, *on the same set of images*. For all the entries in Tab. 3, we train the model with 60K iterations and a batch size of 1024. We see in Tab. 3 that TCIA performs favorably against E-Replay Chaudhry et al. (2019) (+6.1 points) and L2P+ baselines (+8.9 points), exhibiting minimal forgetting. Notably, our model almost closes the gap with the multi-tasking upper-bound with only 1.5 points of performance drop (60.6 *v.s.* 62.1).

Table 3: **Instruction-incremental learning** on COCO Lin et al. (2014) dataset with 5 different textual instructions and same visual input.

| Method | F↓ | Score ↑ |
|---|---|---|
| *Multi-tasking* | | |
| Finetuning | – | 66.8 |
| Adapter | – | 62.1 |
| *Sequential tasks* | | |
| E-Replay | 13.7 | 53.9 |
| L2P+ | 0.47 | 51.1 |
| TCIA | **0.24** | **60.6** |

Table 4: **Application-incremental learning** on 8 applications (*App.*) across 36 datasets (*Data.*).

| Method | Score ↑ | |
|---|---|---|
| | 8 App. | 36 Data. |
| *Multi-tasking* | | |
| Finetuning | 72.7 | |
| Adapter | 63.7 | |
| *Sequential tasks* | | |
| E-Replay | 57.7 | 53.9 |
| L2P+ | 49.3 | 48.2 |
| TCIA | **59.2** | **67.2** |

## 5.4 APPLICATION-INCREMENTAL LEARNING

Lastly, in Tab. Tab. 4, we conduct experiments on the most challenging learning setup in terms of both task diversity as well as task quantity. We curate a list of 8 applications consisting of 36 datasets, detailed in the Tab. 7, and evaluate the model's capacity in learning incrementally across very diverse visual and textual instruction input. We consider two settings: (i) each application is taken as a task with different datasets within that application combined directly (8 *App.* with 8 tasks in total) and (ii) each dataset is taken as a task (36 *Data.* with 36 tasks in total). For all entries in Tab. Tab. 4, we train the model with 400K iterations with a batch size of 1024. Our results with TCIA outperform E-Replay Chaudhry et al. (2019) by a large margin of +1.5 points and +14.3 points on two settings, respectively. This challenging setting validates our method's effectiveness in alleviating forgetting in realistic incremental learning environments. Notably, TCIA with 36 task codes outperforms the multi-tasking adapter baseline by +3.5 points (67.2 *v.s.* 63.7). This suggests that the proposed task codebook not only helps alleviate forgetting but also improves final performance upper-bound.

## 5.5 ABLATION STUDY

In this section, we thoroughly analyze TCIA to assess the impact of each component. To complement the end-task incremental learning score, we introduce an additional metric, $Acc_L$, which directly quantifies the lookup module's performance by measuring the accuracy of selecting the correct task key during inference.

**Task key representation.** We study the impact of different task key representations on codebook lookup accuracy (Tabs. 5i and 5ii) and end-task incremental learning score (Tab. 5iii). We find that using *max-pooling* over the sequence feature yields robust lookup accuracy over different incremental learning setups. This echos studies on text classification Chen (2015); Conneau et al. (2017) finding that max-pooling excels at capturing salient information for tasks like sentiment analysis where the presence of certain keywords is informative. Notably, we find that `[CLS]` and `[EOS]` are complementary-`[CLS]` excels at dataset-incremental learning while `[EOS]` excels at instruction-incremental learning. Although their concatenation offers higher lookup accuracy hence end-task score, it still lags behind from simply using max-pooling. See Tab. 17 for additional results on other setups.

**Task codebook depth.** We incorporate the task codebook into the $l$-th decoder layer, effectively adapting the subsequent $N - l$ layers. We use a GiT decoder with $N = 6$ layers and investigate the impact of task codebook depth by varying $l$ in Tabs. 5i and 5ii. Results indicate that positioning the codebook at the shallowest layer ($l = 0$) performs more robustly against an intermediate placement ($l \geq 1$) when using max-pooling as task key representation. For models with the task codebook placed at deeper layers, the end-task score decline is primarily attributed to low $Acc_L$, suggesting that the model struggles to identify the appropriate task key at inference.

**Update strategy.** We discuss two alternatives to update the task-specific keys described in Section 3.2. As shown in Tab. 6i, employing Exponential Moving Average (EMA) with $m = 0.99$ for task code updates yields a higher performance for both captioning and VQA. In comparison to learning the keys via gradient backpropagation as in L2P approach Wang et al. (2022c), we observe improvements of +2.4 points and +1.1 points on captioning and VQA, respectively.

Table 5: **Ablation study** on different choices of task key representations (*Rep.*) and task codebook depth $l$. **Left:** Task codebook lookup accuracy (Acc$_L$) with the default setup underlined. **Right:** incremental learning score with the optimal $l$. Task key representations including (a) [CLS]; (b) [EOS]; (c) [[CLS];[EOS]]; (d) mean-pooling; (e) max-pooling; (f) [[CLS];max-pooling]; (g) [[CLS];[EOS];max-pooling] (h) CLIP-B Radford et al. (2021) are ablated.

| $l$ | (a) | (b) | (c) | (d) | (e) | (f) | (g) | (h) |
|---|---|---|---|---|---|---|---|---|
| 0 | 45.7 | 93.7 | 95.3 | 49.2 | 94.6 | 87.3 | 95.0 | 89.5 |
| 1 | 46.9 | 92.0 | 96.0 | 62.2 | 95.1 | 83.0 | 94.6 | 89.5 |
| 2 | 66.6 | 91.3 | 95.4 | 66.9 | 96.2 | 96.0 | 95.0 | 89.5 |
| 3 | 83.7 | 92.9 | 95.2 | 86.6 | 94.7 | 90.8 | 96.3 | 89.5 |

(i) **Acc$_L$.** Dataset-incremental learning on VQA datasets.

| $l$ | (a) | (b) | (c) | (d) | (e) | (f) | (g) | (h) |
|---|---|---|---|---|---|---|---|---|
| 0 | 53.6 | 40.2 | 80.5 | 58.9 | 96.9 | 87.6 | 82.6 | 71.3 |
| 1 | 59.7 | 42.2 | 81.4 | 66.9 | 94.3 | 85.9 | 83.1 | 71.3 |
| 2 | 72.1 | 51.5 | 82.7 | 74.1 | 87.0 | 83.3 | 85.0 | 71.3 |
| 3 | 79.3 | 54.8 | 82.1 | 74.6 | 82.8 | 83.2 | 85.6 | 71.3 |

(ii) **Acc$_L$.** Application-incremental learning on 36 datasets.

| Rep. | Data. Cap. | Data. VQA | Inst. | App. |
|---|---|---|---|---|
| (a) | 90.6 | 45.9 | 35.7 | 49.7 |
| (b) | 64.2 | 55.1 | 58.9 | 31.3 |
| (c) | 97.4 | 55.1 | 58.2 | 53.7 |
| (d) | 91.2 | 47.8 | 46.8 | 50.1 |
| (e) | **100.8** | 55.1 | **60.6** | **67.2** |
| (f) | 92.8 | 54.0 | 57.2 | 59.9 |
| (g) | 97.8 | **55.2** | **60.6** | 55.8 |
| (h) | 94.3 | 53.2 | **60.6** | 51.4 |

(iii) **Incremental learning score**. *Data.*, *Inst.*, and *App.* denotes dataset-, instruction-, and application-incremental learning setup, respectively.

Table 6: **Ablation study** on the design of the update strategy (**left**), internal dimensionality (**middle**), and adaptation strategy (**right**) of the task codebook. We report the task score. *+Param.* refers to the ratio of additional parameters introduced by TCIA compared to the entire model *per layer per task*.

| Update | $m$ | Cap. | VQA |
|---|---|---|---|
| Backprop | – | 98.4 | 54.0 |
| EMA | 0.97 | 98.3 | 54.9 |
| EMA | 0.99 | **100.8** | **55.1** |
| EMA | 0.996 | 100.0 | 54.4 |

(i)

| $s$ | +Param. | Cap. | VQA |
|---|---|---|---|
| 128 | 0.1% | 98.0 | 55.0 |
| 256 | 0.2% | **100.8** | **55.1** |
| 512 | 0.4% | 98.0 | 54.3 |

(ii)

| Attn. | MLP | +Param. | Cap. | VQA |
|---|---|---|---|---|
| ✓ | | 0.1% | 96.8 | 54.9 |
| | ✓ | 0.1% | 99.0 | 55.0 |
| ✓ | ✓ | 0.2% | **100.8** | **55.1** |

(iii)

**Internal dimensionality.** As described in Sec. 3.2, TCIA[A] incorporates adapter-based task-specific values consisting of 2-layer MLPs with internal dimensionality $s$. In Tab. 6ii, we assess the impact of this parameter $s$ both in terms of performance and parameter count increase. Our results show that a value of $s = 256$ offers an optimal balance between accuracy and memory consumption. Beyond this point, performance gains start to saturate, while the memory footprint keeps increasing.

**Adaptation strategy.** By default, we use two 2-layer MLPs to adapt the generative model, one after the self-attention (*Attn.*) layer and the other after the MLP layer Houlsby et al. (2019). We explore whether the memory consumption can be further decreased by discarding either one of the two 2-layer MLPs. Results in Tab. 6iii indicate that adapting with two placements achieves optimal results.

## 5.6 ANALYSIS

**Confusion Matrix.** We study the task key selection accuracy (Acc$_L$) further by showcasing the confusion matrix between different datasets. In Fig. 3, we observe that the model can mostly retrieve the correct task-specific key (*i.e.*, 96.9% accuracy). TextVQA Singh et al. (2019) in the OCR-CV application is mostly easily confused with TDIUC Kafle & Kanan (2017) in the VQA application. Fashion-MNIST Xiao et al. (2017) and NotMNIST Bulatov (2011) in the classification application are sometimes confused with COCO Lin et al. (2014) in the detection application and ChartQA Masry et al. (2022) in the OCR-CV application.

**Oracle.** We present in Fig. 4 the performance of TCIA in the presence of an oracle, *i.e.* selecting the ground-truth task key at inference time. This assesses the potential performance gains we could have with better task lookup designs and represents an upper-bound for TCIA. Interestingly, we observe that there are still potential gains by improving the task-specific lookup accuracy: *i.e.* +10.9 points on CIFAR-100 and +6.6 points on 8 applications. On the other hand, there is virtually no gain by having an oracle in VQA dataset incremental learning setup since Acc$_L$ is already very high (94.6%

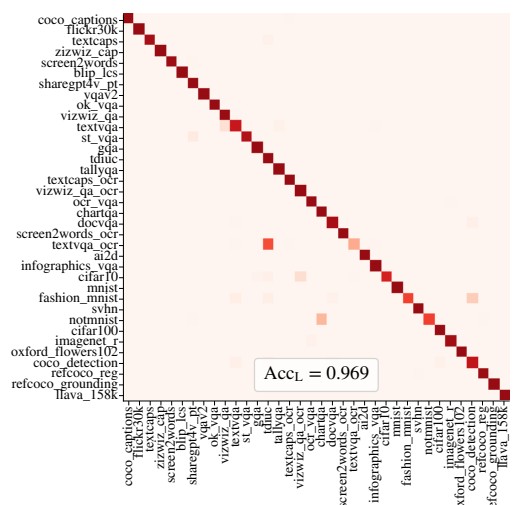

Figure 3: **Confusion matrix** for application-incremental learning across 36 datasets.

Figure 4: **Performance improvements** in terms of incremental learning score when considering an oracle retrieving the correct task at inference time.

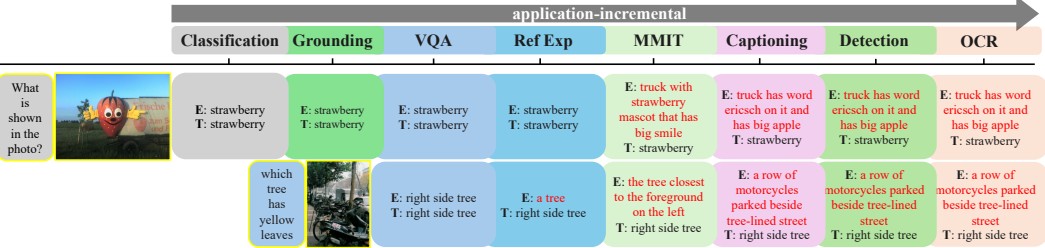

Figure 5: **Qualitative comparison** between TCIA ($T$) and E-Replay Chaudhry et al. (2019) ($E$) with incremental learning on 8 applications.

accuracy). This suggests that further enhancements in task lookup accuracy could lead to substantial performance gains and is a promising direction for future work.

**Qualitative Analysis.** We showcase in Fig. 5 the qualitative comparison of the forgetting behaviour between TCIA and the E-Replay method Chaudhry et al. (2019). Comparatively, the output from E-Replay is more prone to drift from ground truth after subsequent tasks are finished. For example, after being fine-tuned on the MMIT application, E-Replay mistakenly predicts "*truck with strawberry mascot that has big smile*", while TCIA robustly predicts the label name of the image "*strawberry*".

# 6 CONCLUSION

In this paper, we expand the setting of incremental learning to the scope of VLMs. With a high diversity of tasks, datasets and modalities, we model a realistic setting of adapting large multi-modal models for various downstream applications while minimizing forgetting. We introduce a new simple incremental learning method, TCIA, that improves over prior proposed methods and baselines, especially for complex tasks. We extensively evaluate our new incremental learning method on 36 datasets and 8 applications. We hope that our paper will help the community to advance incremental learning research for large multi-modal models.

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

Table 7: **Statistics for 36 datasets and 8 applications**.

| Dataset | Visual Domain | #Train | #Val | #Test | Metric |
|---|---|---|---|---|---|
| *Captioning* | | | | | |
| COCO Lin et al. (2014) | Natural Images | 113287 | 5000 | 5000 | |
| Flickr30K Young et al. (2014) | Natural Images | 29000 | 1014 | 1000 | |
| TextCaps Sidorov et al. (2020) (**w/o OCR**) | Natural Images | 109725 | 15830 | - | |
| VizWiz-Cap Gurari et al. (2018) (**w/o OCR**) | Natural Images | 100575 | 33145 | - | CIDEr |
| Screen2Words Wang et al. (2021) (**w/o OCR**) | UIs | 15743 | 2364 | 4310 | |
| BLIP-LCS Li et al. (2022); Liu et al. (2023) | Natural Images | 446502 | 111626 | - | |
| ShareGPT4V-PT Chen et al. (2023a) | Natural Images | 997520 | 249381 | - | |
| *VQA* | | | | | |
| VQAv2 Goyal et al. (2017) | Natural Images | 594917 | 26261 | 25829 | VQA Acc. |
| OK-VQA Marino et al. (2019) | Natural Images | 8998 | 5033 | - | VQA Acc. |
| VizWiz-QA Gurari et al. (2018) (**w/o OCR**) | Natural Images | 20523 | 4319 | - | VQA Acc. |
| TextVQA Singh et al. (2019) (**w/o OCR**) | Natural Images | 34602 | 5000 | - | VQA Acc. |
| ST-VQA Biten et al. (2019) (**w/o OCR**) | Natural Images | 23446 | 2628 | - | VQA Acc. |
| GQA Hudson & Manning (2019) | Natural Images | 943000 | 132062 | 12578 | VQA Acc. |
| TDIUC Kafle & Kanan (2017) | Natural Images | 1003061 | 50000 | - | VQA Acc. |
| TallyQA Acharya et al. (2019) | Natural Images | 249318 | 38589 | - | EM |
| *OCR-CV* | | | | | |
| TextCaps Sidorov et al. (2020) | Natural Images | 109725 | 15830 | - | CIDEr |
| VizWiz-QA Gurari et al. (2020) | Natural Images | 20523 | 4319 | - | VQA Acc. |
| OCR-VQA Mishra et al. (2019) | Illustrations | 805110 | 31000 | 31000 | EM |
| ChartQA Masry et al. (2022) | Illustrations | 28299 | 1920 | 2500 | RA |
| DocVQA Mathew et al. (2021) | Documents | 39463 | 5349 | 5188 | ANLS |
| Screen2Words Wang et al. (2021) | UIs | 15743 | 2364 | 4310 | CIDEr |
| TextVQA Singh et al. (2019) | Natural Images | 34602 | 5000 | - | VQA Acc. |
| AI2D Kembhavi et al. (2016) | Illustrations | 12293 | 120 | 3088 | EM |
| InfographicsVQA Mathew et al. (2022) | Documents | 23946 | 2801 | 3288 | ANLS |
| *Open-Vocabulary Classification* | | | | | |
| CIFAR10 Krizhevsky et al. (2009) | Natural Images | 50000 | 10000 | - | |
| MNIST LeCun et al. (1998) | Natural Images | 60000 | 10000 | - | |
| Fashion-MNIST Xiao et al. (2017) | Natural Images | 60000 | 10000 | - | |
| SVHN Netzer et al. (2011) | Natural Images | 73257 | 26032 | - | EM |
| notMNIST Bulatov (2011) | Natural Images | 59916 | 14979 | - | |
| CIFAR100 Krizhevsky et al. (2009) | Natural Images | 50000 | 10000 | - | |
| ImageNet-R Hendrycks et al. (2021) | Natural Images | 24000 | 6000 | - | |
| Flowers Nilsback & Zisserman (2008) | Natural Images | 1020 | 1020 | - | |
| *Detection* | | | | | |
| COCO Lin et al. (2014) | Natural Images | 113287 | 5000 | 5000 | AP |
| *Referring Expression Generation* | | | | | |
| RefCOCO Kazemzadeh et al. (2014) | Natural Images | 287604 | 26488 | 30969 | CIDEr |
| *Grounding* | | | | | |
| RefCOCO Kazemzadeh et al. (2014) | Natural Images | 287604 | 26488 | 30969 | IoU |
| *Multi-Modal Instructions* | | | | | |
| LLaVA-158K Liu et al. (2023) | Natural Images | 126169 | 31523 | - | CIDEr |

# A IMPLEMENTATION DETAILS

## A.1 APPLICATIONS

In this section, we detail the definition considered 8 applications and a total of 36 datasets we curated. See Tab. 7 for the statistics and full details.

Table 8: **Instruction template for 8 applications**.

| |
| --- |
| *Captioning* |
| Generate the caption for this image. |
| Generate the caption |
| Generate the alt_text: |
| Describe the image. |
| Caption this image. |
| A photo of |
| Caption |
| A short image caption |
| summarize the given photo |
| detail everything you see in this image |
| Provide a detailed description of this image, including objects, actions, and context. |
| What is going on in the image? |
| what is the main content of this image? |
| what is shown in the photo |
| Briefly describe the content of the image. |
| Use a few words to illustrate what is happening in the picture. |
| *VQA* |
| {Question} |
| *OCR-CV* |
| {*Captioning* or *VQA* Instruction} + <OCR text: ...> |
| *Open-Vocabulary Classification* |
| Identify the primary subject in this image. |
| Name the object in the image. |
| Classify this image |
| classify |
| What do you see here? |
| What category does this image belong to? |
| What is the main object of this image? |
| What is shown in the photo? |
| what is inside this image? |
| what's in this picture |
| *Detection* |
| detect: |
| *Referring Expression Generation* |
| describe the box at {Location}: |
| *Grounding* |
| detect {Referring Expression}: |
| *Multi-Modal Instructions* |
| {Instruction} |

**Captioning** requires the model to provide a concise description of visual content. The following datasets with short captions are considered: COCO Captions Lin et al. (2014) with the split defined in Karpathy & Fei-Fei (2015), Flickr30K Young et al. (2014), TextCaps Sidorov et al. (2020), VizWiz-Cap Gurari et al. (2020), Screen2Words Wang et al. (2021), and BLIP-LCS Li et al. (2022). To account for the increasing length and quality of captions generated by multi-modal instruction-following models, the ShareGPT4V-PT Chen et al. (2023a) dataset is also included, which features significantly longer and more descriptive captions.

**Visual Question Answering (VQA)** aims at generating textual answers based on both a textual question and visual context. For our benchmark, we use datasets where the visual context consists of natural images Chen et al. (2023b): VQAv2 Goyal et al. (2017), OKVQA Marino et al. (2019), TextVQA Singh et al. (2019), VizWiz-QA Gurari et al. (2018), ST-VQA Biten et al. (2019), GQA Hudson & Manning (2019), TDIUC Kafle & Kanan (2017), and TallyQA Acharya et al. (2019).

**OCR-enhanced captioning and VQA (OCR-CV)** leverage the output of an upstream OCR model to solve captioning and VQA applications. We consider the following datasets: TextCaps Sidorov et al. (2020), Screen2Words Wang et al. (2021), VizWiz-QAGurari et al. (2018), OCR-VQA Mishra et al. (2019), ChartQA Masry et al. (2022), DocVQA Mathew et al. (2021), TextVQA Singh et al. (2019), AI2D Kembhavi et al. (2016) and InfographicsVQA Mathew et al. (2022). In these datasets, OCR text tokens are extracted by an upstream OCR system Chen et al. (2023b) and provided as additional text inputs.

**Open-vocabulary classification** categorizes object-centric images into their corresponding classes. In our benchmark, we redefine classification as a generative task. Instead of training a set of classifier weights Chaudhry et al. (2019); Wang et al. (2022c), we use a generative model with open-world vocabulary to directly generate class names Wang et al. (2022a). The considered datasets are CIFAR10 Krizhevsky et al. (2009), CIFAR100 Krizhevsky et al. (2009), MNIST LeCun et al. (1998), notMNIST Bulatov (2011), FashionMNIST Xiao et al. (2017), SVHN Netzer et al. (2011), ImageNet-R Wang et al. (2022b), and Flowers Nilsback & Zisserman (2008).

**Spatial recognition** assesses the model's ability to comprehend and generate output based on spatial coordinates of the image, typically represented as bounding boxes. Following the approach of Pix2Seq Chen et al. (2021; 2023c;b), we directly incorporate discretized bounding boxes into the text input by concatenating them with the relevant object or concept. We evaluate the model for three distinct applications: i) Detection, where the bounding boxes for *all* objects are regarded as the target $\mathcal{Y}$, ii) Referring Expression Generation, where the bounding box for a *specific* object is included in the instruction $\mathcal{I}$, iii) Grounding, where the bounding box for a *specific* object is present in target $\mathcal{Y}$. All three of these applications are performed on the COCO Lin et al. (2014) dataset.

**Multi-modal instruction tuning** considers more complex textual $\mathcal{I}$ inputs, including but not limited to conversational-style QA, detailed descriptions, and complex reasoning. They can be seen as a super-set of all previously considered applications. For this purpose, we employ the 158K supervised fine-tuning dataset from LLaVA Liu et al. (2023) derived from COCO Lin et al. (2014) dataset.

## A.2 DATASETS

Here we specify the details of 36 datasets used in our experiments. Unless specified otherwise as below, we use default train and validation sets.

**BLIP-LCS Li et al. (2022) & ShareGPT4V-PT Chen et al. (2023a).** We use 80% of the original data as the training set and the rest of 20% as the validation set.

**TDIUC Kafle & Kanan (2017).** We use 10% of the original validation set as the validation set here.

**OCR-VQA Mishra et al. (2019).** We use 15% of the original validation set as the validation set here and another 15% as the test set.

**RefCOCO Kazemzadeh et al. (2014).** We count the object numbers since the evaluation is instance-wise for referring expression generation and grounding.

**LLaVA-158K Liu et al. (2023).** We use 158K unique language-image instruction-following samples that collected in LLaVA Liu et al. (2023). The instructions are much more diverse comparing with ones from other applications, including conversation, detailed captioning, and complex reasoning. The images are sampled from COCO Lin et al. (2014) and the answers are taken via interacting with language-only GPT-4. We use 80% of the original data as the training set and the rest of 20% as the validation set.

A.3 PRE-PROCESSING

In this section, we detail necessary pre-processing steps for reproduction purposes. By default, we resize the image with a ratio uniformly sampled from $[0.75, 1.25]$ and pad the image's shorter side. See Tab. 8 for the instruction template we use for different applications. We tokenize the textual instructions and targets using a BERT tokenizer Devlin et al. (2018) with a vocabulary size of 30522.

**Captioning.** We manually append an instruction randomly sampled from a pool detailed in Tab. 8. We directly use the image caption as the target. For datasets with multiple captions (*e.g.*, COCO Captions Lin et al. (2014)), we randomly sample one.

**VQA.** We directly use the question as the instruction and the answer as the target from the original dataset without any other extra processing. For datasets with multiple captions (*e.g.*, VQAv2 Goyal et al. (2017)), we randomly sample one.

**OCR-CV.** For visual input, we do not scale the image while resizing to make sure the entire image is visible to the model. We follow the pre-processing steps as Chen et al. (2023b) using an upstream OCR system, GCP Vision API, to extract the potential OCR texts in the image. The extracted OCR texts are then appended in the format of "*<OCR text: ...>*" after the original captioning or VQA instructions.

**Open-vocabulary classification.** We manually append an instruction randomly sampled from a pool detailed in Tab. 8. For the target, we directly use the class label name if the mapping exists. We use "*digit {Class Label}*" for MNIST LeCun et al. (1998) and SVHN Netzer et al. (2011), "*letter {Class Label}*" for notMNIST Bulatov (2011).

**Detection.** For visual input, we resize the image with a ratio uniformly sampled from $[0.3, 2.0]$ and randomly flip the image. We simply use "*detect:*" as the instruction. For the target, we rescale the range of original bounding boxes in COCO Lin et al. (2014) from 1.0 to 999 (int) and concatenate 4 coordinates directly with the instance label. Different instances are directly concatenated with "*and*". An exemplar target looks like "*10 99 323 675 cat and 127 346 894 997 dog*". We do not use the augmentation trick proposed in Chen et al. (2021).

**Referring expression generation.** The pre-processing for visual input follows the detection task. For each instance, we use "describe the box at *{*Location*}*:" as the instruction and its referring expression as the target.

**Grounding.** The pre-processing for visual input follows the detection task. For each instance, we use "detect *{*Referring Expression*}*:" as the instruction and its location as the target. The formatting of the bounding boxes follows the detection task.

**Multi-modal instruction tuning.** We directly use human instructions as the instruction and the GPT-4 OpenAI (2023)'s output as the target.

Table 9: **Dataset-incremental learning for captioning and VQA** with different task order shuffling seed.

| Seed | 1 | | 2 | | 3 | | 4 | | 5 | | Mean | | Variance | |
|---|---|---|---|---|---|---|---|---|---|---|---|---|---|---|
| | F↓ | Score↑ | F↓ | Score↑ | F↓ | Score↑ | F↓ | Score↑ | F↓ | Score↑ | F↓ | Score↑ | F | Score |
| captioning | 0.47 | 100.8 | 0.04 | 99.8 | 0.32 | 100.6 | 0.59 | 96.2 | 0.09 | 96.1 | **0.30** | **98.7** | 0.04 | 4.44 |
| VQA | 0.54 | 55.1 | 0.77 | 54.9 | 54.9 | 0.03 | 54.8 | 0.71 | 54.4 | 0.09 | **0.43** | **54.8** | 0.10 | 0.05 |

Table 10: **Dataset-incremental learning for the captioning application.** Detailed results on each dataset for single-tasking (ST), multi-tasking (MT) and sequential tasks (Seq) are reported. All results are without OCR in the instruction.

| Dataset | ST | | | | MT | | Seq | | | | | | |
|---|---|---|---|---|---|---|---|---|---|---|---|---|---|
| | Fine-tuning | | Adapter | | Fine-tuning | Adapter | Fine-tuning | Adapter | E-Replay | L2P+TCIA[P] | TCIA[Pr] | TCIA[A] | |
| | Val | Test | Val | Test | Val | Val | Val | Val | Val | Val | Val | Val | Val |
| COCO | 139.8 | 141.7 | 134.2 | 136.1 | 135.6 | 131.3 | 102.1 | 104.7 | 108.0 | 121.4 | 125.9 | 118.6 | 126.1 |
| Flickr30K | 102.6 | 100.9 | 97.8 | 98.0 | 102.4 | 93.9 | 97.5 | 95.0 | 96.4 | 92.5 | 92.1 | 87.6 | 91.2 |
| TextCaps | 100.9 | - | 93.4 | - | 105.9 | 88.7 | 61.3 | 57.0 | 74.1 | 85.0 | 84.8 | 80.3 | 86.8 |
| VizWiz-Cap | 101..7 | - | 92.5 | - | 101.3 | 91.6 | 57.3 | 58.9 | 71.2 | 84.5 | 84.0 | 81.9 | 88.0 |
| Screen2Words | 85.3 | 84.8 | 78.1 | 79.3 | 70.9 | 62.8 | 46.3 | 29.1 | 66.0 | 61.5 | 60.6 | 70.6 | 73.2 |
| BLIP-LCS | 133.8 | - | 122.6 | - | 136.4 | 117.8 | 71.4 | 58.1 | 94.9 | 108.8 | 108.7 | 104.8 | 120.9 |
| ShareGPT4V-PT | 185.8 | - | 114.6 | - | 138.2 | 92.3 | 0.0 | 0.0 | 30.2 | 28.4 | 0.0 | 73.0 | 119.1 |
| **Avg** | **121.4** | - | **104.7** | - | **112.9** | **96.9** | **62.3** | **57.5** | **77.3** | **83.2** | **79.4** | **88.1** | **100.8** |

## A.4 METRICS

For task $t$ sampled from task mixture $\mathbf{T}$, let $\mathcal{A}_{t,i}$ be the evaluation result under task $t$'s metric after training task $i$ for any $i \geq t$. We report the average correctness *score* after all the tasks are trained:

$$\text{Score} = \frac{1}{T} \sum_i \mathcal{A}_{i,T}. \qquad (1)$$

The *forgetting* metric Chaudhry et al. (2018) (denoted by F in our experiments) is computed as

$$F = \frac{1}{T} \sum_i \max(\mathcal{A}_{i,\geq i}) - \mathcal{A}_{i,T} \qquad (2)$$

where $\max(\mathcal{A}_{i,\geq i})$ is the peak evaluation result for task $i$ after when it is trained.

**Application-Incremental Learning.** To equalize the importance of each application, we first average the evaluation results across datasets within each application and then average the results across applications.

## A.5 TASK ORDER

For all experiments conducted with sequential tasks $\mathbf{T}$, we shuffle it with a fixed seed 1, *i.e.*, `import random; random.Random(1).shuffle(T)`. We include results with seeds $2, 3, 4, 5$ for captioning and VQA in Tab. 9. We report the results of dataset-incremental learning for captioning and VQA on 5 different shuffling seeds. Even though the task order is an important parameter because of the similarity between different tasks, the result indicates that our method is robust to different task orders.

## B ADDITIONAL RESULTS

### B.1 DATASET-INCREMENTAL LEARNING.

We detail the evaluation results for dataset-incremental learning on captioning, VQA, OCR-CV, and open-vocabulary classification in Tab 10, Tab. 11, Tab. 12 and Tab. 13, respectively. Results on

Table 11: **Dataset-incremental learning for the VQA application.** Detailed results on each dataset for single-tasking (ST), multi-tasking (MT) and sequential tasks (Seq) are reported. All results are without OCR in the instruction.

| Dataset | ST Fine-tuning Val | ST Fine-tuning Test | ST Adapter Val | ST Adapter Test | MT Fine-tuning Val | MT Adapter Val | Seq Fine-tuning Val | Seq Adapter Val | Seq E-Replay Val | Seq L2P Val | Seq TCIA$^P$ Val | Seq TCIA$^{Pr}$ Val | Seq TCIA$^A$ Val |
|---|---|---|---|---|---|---|---|---|---|---|---|---|---|
| VQAv2 | 75.1 | 74.6 | 74.6 | 74.5 | 74.8 | 72.1 | 62.4 | 55.1 | 72.1 | 51.2 | 65.5 | 69.8 | 72.8 |
| OK-VQA | 38.1 | - | 41.9 | - | 49.2 | 47.7 | 22.8 | 15.5 | 35.3 | 35.8 | 34.3 | 39.7 | 40.4 |
| VizWiz-QA | 56.9 | - | 56.3 | - | 57.6 | 55.3 | 56.5 | 55.2 | 54.3 | 52.9 | 53.3 | 53.5 | 56.4 |
| TextVQA | 32.6 | - | 30.7 | - | 33.9 | 30.3 | 21.5 | 20.0 | 22.4 | 26.6 | 25.7 | 28.5 | 29.8 |
| ST-VQA | 27.6 | - | 25.9 | - | 31.6 | 27.2 | 21.3 | 17.9 | 23.8 | 21.3 | 20.6 | 22.3 | 24.6 |
| GQA | 66.7 | 57.9 | 63.6 | 56.2 | 67.3 | 63.6 | 46.9 | 36.7 | 61.7 | 59.2 | 54.5 | 56.3 | 59.1 |
| TDIUC | 91.2 | - | 91.0 | - | 92.7 | 90.9 | 60.4 | 53.4 | 90.6 | 80.3 | 89.1 | 90.4 | 90.8 |
| TallyQA | 65.6 | - | 67.0 | - | 67.7 | 67.3 | 47.5 | 37.4 | 61.8 | 64.0 | 63.2 | 65.1 | 66.7 |
| **Avg** | **56.7** | **-** | **56.4** | **-** | **59.4** | **56.8** | **42.4** | **36.4** | **52.7** | **48.9** | **50.7** | **53.2** | **55.1** |

Table 12: **Dataset-incremental learning for the OCR-CV application.** Detailed results on each dataset for single-tasking (ST), multi-tasking (MT) and sequential tasks (Seq) are reported. All results are with OCR in the instruction.

| Dataset | ST Fine-tuning Val | ST Fine-tuning Test | ST Adapter Val | ST Adapter Test | MT Fine-tuning Val | MT Adapter Val | Seq Fine-tuning Val | Seq Adapter Val | Seq E-Replay Val | Seq L2P+ Val | Seq TCIA$^P$ Val | Seq TCIA$^{Pr}$ Val | Seq TCIA$^A$ Val |
|---|---|---|---|---|---|---|---|---|---|---|---|---|---|
| TextCaps | 101.0 | - | 99.8 | - | 104.7 | 88.9 | 13.0 | 5.1 | 83.5 | 81.5 | 81.9 | 80.7 | 95.7 |
| VizWiz-QA | 56.9 | - | 55.8 | - | 57.1 | 53.0 | 16.0 | 7.0 | 42.9 | 52.3 | 51.1 | 53.3 | 56.4 |
| OCR-VQA | 70.1 | 70.3 | 62.6 | 63.0 | 61.5 | 56.8 | 61.0 | 58.2 | 61.0 | 48.0 | 46.6 | 51.5 | 60.3 |
| ChartQA | 18.3 | 17.4 | 19.8 | 19.5 | 19.2 | 18.4 | 10.7 | 3.1 | 12.9 | 12.7 | 12.6 | 14.3 | 18.9 |
| DocVQA | 16.4 | 4.2 | 20.3 | 6.6 | 7.2 | 4.2 | 1.9 | 1.0 | 3.7 | 4.5 | 3.2 | 9.1 | 14.5 |
| Screen2Words | 98.0 | 97.6 | 91.9 | 95.5 | 29.9 | 34.6 | 8.4 | 3.8 | 23.1 | 67.5 | 67.5 | 74.6 | 85.6 |
| TextVQA | 36.7 | - | 34.5 | - | 41.5 | 38.3 | 18.0 | 7.7 | 27.2 | 17.1 | 26.2 | 29.6 | 35.3 |
| AI2D | 35.0 | 30.6 | 40.0 | 35.3 | 37.5 | 40.0 | 3.3 | 0.8 | 15.8 | 28.3 | 24.2 | 31.7 | 37.5 |
| InfographicsVQA | 9.4 | 4.0 | 11.1 | 6.6 | 4.8 | 2.0 | 0.6 | 0.0 | 2.3 | 6.6 | 4.3 | 7.6 | 6.9 |
| **Avg** | **49.1** | **-** | **48.4** | **-** | **40.4** | **37.4** | **14.8** | **9.6** | **30.3** | **35.4** | **35.3** | **39.2** | **45.7** |

Table 13: **Dataset-incremental learning for the open-vocabulary classification application.** Detailed results on each dataset for single-tasking (ST), multi-tasking (MT) and sequential tasks (Seq) are reported.

| Dataset | ST Fine-tuning Val | ST Fine-tuning Test | ST Adapter Val | ST Adapter Test | MT Fine-tuning Val | MT Adapter Val | Seq Fine-tuning Val | Seq Adapter Val | Seq E-Replay Val | Seq L2P+ Val | Seq TCIA$^P$ Val | Seq TCIA$^{Pr}$ Val | Seq TCIA$^A$ Val |
|---|---|---|---|---|---|---|---|---|---|---|---|---|---|
| CIFAR10 | 99.4 | - | 98.6 | - | 98.7 | 96.9 | 73.7 | 2.2 | 95.8 | 97.4 | 97.6 | 97.8 | 98.3 |
| MNIST | 99.3 | - | 99.4 | - | 99.7 | 99.3 | 0.0 | 0.0 | 97.5 | 98.8 | 98.7 | 99.0 | 99.2 |
| Fashion-MNIST | 96.2 | - | 94.0 | - | 96.2 | 93.5 | 95.9 | 93.2 | 95.8 | 90.9 | 91.2 | 91.5 | 93.4 |
| SVHN | 98.0 | - | 90.8 | - | 98.1 | 89.2 | 0.0 | 0.0 | 88.2 | 81.6 | 81.5 | 84.8 | 89.3 |
| notMNIST | 97.9 | - | 95.9 | - | 97.8 | 95.1 | 0.0 | 0.0 | 62.4 | 92.6 | 92.4 | 93.7 | 95.1 |
| CIFAR100 | 94.1 | - | 89.6 | - | 93.1 | 87.6 | 18.3 | 0.1 | 50.7 | 82.7 | 85.3 | 86.4 | 88.0 |
| ImageNet-R | 93.5 | - | 93.3 | - | 93.5 | 92.7 | 3.0 | 0.2 | 19.3 | 90.7 | 90.6 | 92.1 | 92.9 |
| Flowers | 98.0 | - | 96.3 | - | 98.2 | 96.2 | 2.2 | 0.0 | 26.5 | 91.9 | 91.2 | 94.7 | 95.9 |
| **Avg** | **97.1** | **-** | **94.7** | **-** | **96.9** | **93.8** | **24.2** | **11.9** | **67.0** | **90.8** | **90.9** | **92.5** | **94.0** |

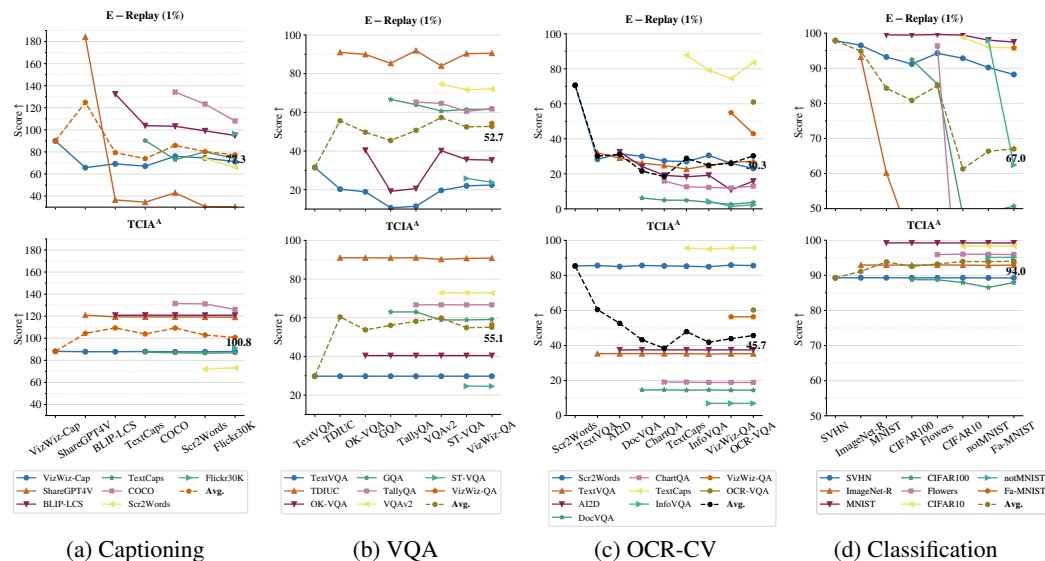

Figure 6: **Dataset-incremental learning** with incremental learning scores over training steps. We compare TCIA[A] (**bottom**) to the E-Replay Chaudhry et al. (2019) method (**top**) with buffer size being 1% of the entire training data. TCIA[A] exhibits minimal forgetting, hence better final accuracy (*Avg.*).

Table 14: **Instruction-incremental learning.** Detailed results on each dataset for single-tasking (ST), multi-tasking (MT), and sequential tasks (Seq) are reported.

| Dataset | ST | | MT | | Seq | | | Metrics |
|---|---|---|---|---|---|---|---|---|
| | Fine-tuning | Adapter | Fine-tuning | Adapter | E-Replay | L2P+ | TCIA[A] | |
| | Val | Val | Val | Val | Val | Val | Val | |
| *Captioning* COCO | 139.8 | 134.2 | 132.8 | 130.5 | 114.2 | 132.3 | 130.2 | CIDEr |
| *VQA* OK-VQA | 38.1 | 41.9 | 38.9 | 41.2 | 35.0 | 38.1 | 39.8 | VQA Acc. |
| *Detection* COCO | 20.1 | 4.3 | 12.9 | 2.80 | 11.2 | 0.7 | 3.3 | AP |
| *Referring Expression Generation* RefCOCO | 92.7 | 76.8 | 86.1 | 89.0 | 62.9 | 63.0 | 85.0 | CIDEr |
| *Grounding* RefCOCO | 66.1 | 44.7 | 61.8 | 47.0 | 46.0 | 21.1 | 44.8 | IoU |
| **Avg** | **71.4** | **60.4** | **66.8** | **62.1** | **53.9** | **51.1** | **60.6** | Score |

detection, referring expression generation, and grounding are with an image resolution of 352, which is chosen as the multiple of 32 closest to 336. All the other results are with an image resolution of 224. The 4 dataset-incremental learning settings on captioning, VQA, OCR-CV, and open-vocabulary classification are trained with 60K, 60K, 45K, and 20K iterations, respectively. The training is conducted with TPU v4 chips.

**Incremental learning score curve.** Additionally, we visualize the incremental learning scores over training steps in Fig. 6. We see that the performance curves of TCIA[A] stay flat as training proceeds, while those of the E-Replay method undergo a noticeable drop, indicating catastrophic forgetting.

Table 15: **Class-incremental learning on varying class splits of CIFAR100 Krizhevsky et al. (2009); Wang et al. (2022c).** *Rep.* denotes the task key representation we use to learn task-specific keys. We specify the type of head used to perform classification: *Classes* refer to a standard unfrozen linear classification head. *Text Vocabulary* is a frozen linear layer outputting word tokens logits. In this case, classification is performed by auto-regressively generating the class label name. All the numbers are run by us.

| Method | Model | Rep. | Head | 5 splits | | 10 splits | | 20 splits | | 100 splits | |
|---|---|---|---|---|---|---|---|---|---|---|---|
| | | | | F↓ | Score ↑ | F↓ | Score ↑ | F↓ | Score ↑ | F↓ | Score ↑ |
| Upper-bound | ViT-B | – | Classes | – | 90.9 | – | 90.9 | – | 90.9 | – | 90.9 |
| Upper-bound | GiT-L | – | Classes | – | 94.1 | – | 94.1 | – | 94.1 | – | 94.1 |
| L2P | ViT-B | ViT-B | Classes | 87.9 | 24.0 | 7.82 | 83.1 | 10.4 | 80.6 | 84.5 | 3.82 |
| E-Replay | GiT-L | – | Classes | 5.37 | 85.3 | **6.12** | 83.9 | 7.44 | 82.8 | 76.2 | 20.4 |
| TCIA | GiT-L | max-pooling | Classes | 4.60 | 86.5 | 6.41 | **87.6** | 7.34 | 83.1 | 4.50 | 11.2 |
| TCIA | GiT-L | max-pooling | Text Vocabulary | 6.18 | 84.3 | 10.2 | 81.7 | 4.11 | 88.0 | 1.42 | 97.0 |
| TCIA | GiT-L | CLIP-B | Text Vocabulary | **2.32** | 86.4 | 11.0 | 83.4 | 4.73 | 87.2 | 0.87 | 96.7 |
| TCIA | GiT-L | ViT-B | Text Vocabulary | 2.54 | **90.3** | 9.16 | 87.0 | **2.18** | **94.3** | **0.75** | **98.1** |

## B.2 CLASS-INCREMENTAL LEARNING.

**Class-incremental learning with varying splits on CIFAR100.** Here, we explore different ways of splitting CIFAR100 for class-incremental learning. We vary the number of splits from 5 to 100 and report results in Tab. 15. Note that having more splits results in a lower number of classes per split, which makes incremental learning more challenging. In fact, with 100 splits, there is only one class per task. Note that this scenario may not be deemed very practical, but we use it to investigate the performance of our method in an extreme setting.

We observe in Tab. 15 that TCIA is robust to the number of considered splits. Both L2P Wang et al. (2022c) and E-Replay Chaudhry et al. (2019) fail when trained incrementally class per class (*i.e.* 100 splits). We find that this is mainly because these baselines use a standard *unfrozen* classification head, which cannot be trained properly by seeing only one class at a time. In contrast, because we build on a generative image-to-text model, we perform classification by auto-regressively generating class names. This way, our default setting is to use a *frozen* text vocabulary classifier as the final head, which outputs logits for word tokens. This alleviates the issue of having to learn a single classification head vector at a time and explains why TCIA obtains a good performance in the 100 splits setting. In fact, we get poor performance similar to L2P and E-Replay when using a classification head instead of a vocabulary head in the TCIA framework. Finally, we see in Tab. 15 that building task-specific keys from additional models CLIP-B Radford et al. (2021) or ViT-B Dosovitskiy et al. (2020) works better than our default design of using inner GiT-L decoder features. Overall, our analysis in Tab. 15 shows the benefit of leveraging generative auto-regressive models for extreme class-incremental learning such as learning one class at a time.

**Accuracy score.** We visualize the incremental learning scores of E-Replay and TCIA across the training steps in Fig. 7.

## B.3 INSTRUCTION-INCREMENTAL LEARNING.

We detail the evaluation results for instruction-incremental learning from Tab. 4 in Tab. 14. Results are with an image resolution of 352 with 60K iterations. The training is conducted with TPU v4 chips.

## B.4 APPLICATION-INCREMENTAL LEARNING.

We detail the evaluation results for application-incremental learning from Tab. 5 in in Tab. 16. Results are with an image resolution of 352 with 400K iterations. The training is conducted with TPU v4 chips.

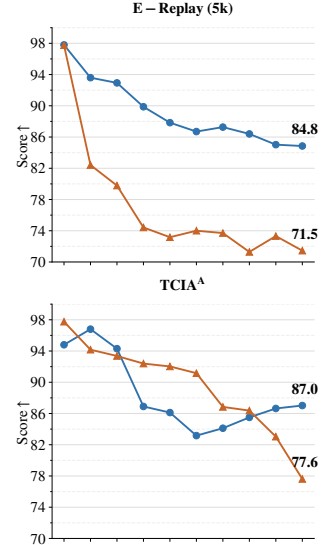

Table 16: **Application-incremental learning.** Detailed results on each dataset for multi-tasking (MT) and sequential tasks (Seq) are reported. For sequential tasks, two setup on 8 applications (*App.*) and 36 datasets (*Data.*) are reported.

| Dataset | MT | | Seq (8 App.) | | | Seq (36 Data.) | | | Metrics |
|---|---|---|---|---|---|---|---|---|---|
| | Fine-tuning | Adapter | E-Replay | L2P+ | TCIA$^A$ | E-Replay | L2P+ | TCIA$^A$ | |
| | Val | Val | Val | Val | Val | Val | Val | Val | |
| *Captioning* | | | | | | | | | |
| COCO | 121.3 | 118.0 | 103.5 | 90.5 | 27.1 | 95.3 | 42.2 | 119.3 | |
| Flickr30K | 79.0 | 89.9 | 63.8 | 80.3 | 26.7 | 54.8 | 63.9 | 85.1 | |
| TextCaps (**w/o OCR**) | 102.5 | 80.1 | 90.8 | 62.0 | 55.7 | 73.3 | 52.3 | 76.3 | |
| VizWiz-Cap (**w/o OCR**) | 93.2 | 78.4 | 67.1 | 21.6 | 53.3 | 63.7 | 57.2 | 79.4 | CIDEr |
| Screen2Words (**w/o OCR**) | 36.7 | 29.6 | 25.7 | 17.0 | 19.6 | 37.6 | 11.1 | 65.2 | |
| BLIP-LCS | 124.4 | 96.6 | 77.5 | 88.8 | 103.0 | 69.4 | 93.1 | 105.2 | |
| ShareGPT4V-PT | 156.7 | 106.7 | 89.9 | 71.6 | 137.4 | 50.4 | 75.5 | 138.6 | |
| *VQA* | | | | | | | | | |
| VQAv2 | 73.0 | 69.2 | 65.3 | 62.9 | 71.0 | 70.4 | 47.9 | 67.9 | |
| OK-VQA | 46.2 | 45.1 | 31.2 | 21.9 | 43.1 | 44.1 | 29.4 | 40.4 | |
| VizWiz-QA (**w/o OCR**) | 54.5 | 48.7 | 50.7 | 20.8 | 40.2 | 43.4 | 44.24 | 51.6 | |
| TextVQA (**w/o OCR**) | 47.3 | 30.9 | 41.5 | 22.9 | 29.4 | 31.8 | 24.6 | 29.4 | VQA Acc. |
| ST-VQA (**w/o OCR**) | 42.2 | 27.5 | 35.1 | 19.4 | 26.7 | 27.7 | 20.5 | 24.7 | |
| GQA | 63.9 | 63.2 | 54.7 | 57.2 | 63.6 | 59.1 | 42.4 | 58.1 | |
| TDIUC | 46.2 | 90.2 | 88.6 | 88.0 | 90.6 | 88.9 | 86.3 | 88.8 | |
| TallyQA | 54.5 | 67.1 | 59.5 | 61.1 | 68.2 | 61.3 | 65.3 | 64.4 | EM |
| *OCR-CV* | | | | | | | | | |
| TextCaps | 47.3 | 96.2 | 100.2 | 40.5 | 95.6 | 75.7 | 54.8 | 85.5 | CIDEr |
| VizWiz-QA | 42.2 | 49.9 | 51.6 | 14.5 | 47.0 | 45.5 | 46.6 | 53.3 | VQA Acc. |
| OCR-VQA | 63.9 | 55.9 | 58.4 | 43.0 | 58.7 | 47.5 | 46.5 | 59.5 | EM |
| ChartQA | 91.1 | 21.5 | 26.1 | 12.4 | 25.9 | 14.4 | 14.0 | 20.0 | RA |
| DocVQA | 66.5 | 22.3 | 8.9 | 27.9 | 26.5 | 11.1 | 9.1 | 19.1 | ANLS |
| Screen2Words | 105.3 | 31.7 | 26.9 | 20.5 | 39.3 | 39.4 | 6.5 | 71.6 | CIDEr |
| TextVQA | 54.9 | 40.7 | 48.1 | 21.2 | 42.3 | 34.0 | 23.8 | 33.6 | VQA Acc. |
| AI2D | 40.8 | 35.0 | 42.5 | 2.5 | 36.7 | 8.3 | 24.2 | 35.0 | EM |
| InfographicsVQA | 12.8 | 10.4 | 11.3 | 5.6 | 10.5 | 6.0 | 3.4 | 6.9 | ANLS |
| *Open-Vocabulary Classification* | | | | | | | | | |
| CIFAR10 | 98.3 | 94.5 | 90.2 | 94.0 | 94.5 | 97.4 | 17.9 | 96.9 | |
| MNIST | 99.7 | 99.1 | 98.3 | 98.5 | 99.3 | 99.4 | 99.2 | 99.2 | |
| Fashion-MNIST | 96.1 | 92.4 | 89.3 | 90.1 | 93.2 | 92.0 | 90.2 | 93.4 | |
| SVHN | 97.9 | 88.3 | 92.0 | 82.6 | 89.5 | 93.8 | 85.5 | 91.0 | |
| notMNIST | 98.1 | 94.4 | 95.0 | 92.3 | 94.8 | 91.9 | 89.2 | 95.2 | EM |
| CIFAR100 | 91.7 | 82.4 | 59.8 | 78.9 | 83.1 | 64.0 | 75.5 | 83.8 | |
| ImageNet-R | 90.5 | 90.3 | 37.6 | 85.1 | 55.8 | 36.2 | 86.8 | 91.6 | |
| Flowers | 98.0 | 94.5 | 28.5 | 87.4 | 95.4 | 24.6 | 87.5 | 96.1 | |
| *Detection* | | | | | | | | | |
| COCO | 14.1 | 1.2 | 8.9 | 0.3 | 3.2 | 10.9 | 0.3 | 3.5 | AP |
| *Referring Expression Generation* | | | | | | | | | |
| RefCOCO | 97.6 | 73.7 | 55.3 | 60.4 | 74.7 | 61.4 | 67.1 | 81.2 | CIDEr |
| *Grounding* | | | | | | | | | |
| RefCOCO | 65.9 | 31.3 | 41.8 | 20.1 | 47.0 | 50.8 | 20.8 | 47.1 | IoU |
| *Multi-Modal Instructions* | | | | | | | | | |
| LLaVA-158K | 59.3 | 46.8 | 43.9 | 30.4 | 61.8 | 65.1 | 30.8 | 61.9 | CIDEr |
| **Avg** | **72.7** | **63.7** | **57.7** | **49.3** | **59.2** | **53.9** | **48.2** | **67.2** | Score |

Table 17: **Ablation study** on different choices of sequence representation and task codebook depth $l$ with task codebook lookup accuracy ($\text{Acc}_L$). Task key representations including (a) [CLS]; (b) [EOS]; (c) [[CLS];[EOS]]; (d) mean-pooling; (e) max-pooling; (f) [[CLS];max-pooling]; (g) [[CLS];[EOS];max-pooling] (h) CLIP-B Radford et al. (2021) are ablated. The default setup is underlined.

| $l$ | (a) | (b) | (c) | (d) | (e) | (f) | (g) | (h) |
|---|---|---|---|---|---|---|---|---|
| 0 | 89.2 | 28.2 | 93.4 | 89.2 | 96.1 | 91.0 | 92.8 | 85.3 |
| 1 | 88.7 | 36.4 | 89.4 | 90.6 | 94.1 | 92.0 | 93.9 | 85.3 |
| 2 | 88.4 | 66.9 | 89.6 | 92.2 | 93.9 | 92.2 | 93.4 | 85.3 |
| 3 | 90.3 | 53.8 | 88.4 | 92.7 | 87.1 | 92.7 | 91.2 | 85.3 |

(i) Dataset-incremental learning on captioning datasets.

| $l$ | (a) | (b) | (c) | (d) | (e) | (f) | (g) | (h) |
|---|---|---|---|---|---|---|---|---|
| 0 | 89.5 | 23.8 | 89.5 | 89.9 | 99.4 | 99.8 | 99.7 | 88.1 |
| 1 | 89.9 | 48.9 | 90.7 | 94.3 | 98.8 | 99.3 | 97.4 | 88.1 |
| 2 | 94.2 | 76.9 | 94.5 | 93.1 | 98.0 | 93.8 | 97.1 | 88.1 |
| 3 | 95.2 | 84.6 | 95.5 | 93.7 | 96.7 | 89.7 | 92.3 | 88.1 |

(ii) Dataset-incremental learning on OCR-CV datasets.

| $l$ | (a) | (b) | (c) | (d) | (e) | (f) | (g) | (h) |
|---|---|---|---|---|---|---|---|---|
| 0 | 99.5 | 12.4 | 97.3 | 99.9 | 99.9 | 99.9 | 98.4 | 99.7 |
| 1 | 99.5 | 17.0 | 97.6 | 99.9 | 93.8 | 99.8 | 97.6 | 99.7 |
| 2 | 98.4 | 55.4 | 99.1 | 99.9 | 95.3 | 99.7 | 99.4 | 99.7 |
| 3 | 98.2 | 61.7 | 95.9 | 99.9 | 95.8 | 99.7 | 99.4 | 99.7 |

(iii) Dataset-incremental learning on classification datasets.

| $l$ | (a) | (b) | (c) | (d) | (e) | (f) | (g) | (h) |
|---|---|---|---|---|---|---|---|---|
| 0 | 42.5 | 10.6 | 42.4 | 72.8 | 84.0 | 54.7 | 56.3 | 84.7 |
| 1 | 40.7 | 16.7 | 39.6 | 73.1 | 65.8 | 54.7 | 57.1 | 84.7 |
| 2 | 42.5 | 14.0 | 45.9 | 78.9 | 60.5 | 57.6 | 60.1 | 84.7 |
| 3 | 46.3 | 11.0 | 42.2 | 82.7 | 71.5 | 56.6 | 53.6 | 84.7 |

(iv) Class-incremental learning on 10 splits of the CIFAR100 Krizhevsky et al. (2009) dataset.

| $l$ | (a) | (b) | (c) | (d) | (e) | (f) | (g) | (h) |
|---|---|---|---|---|---|---|---|---|
| 0 | 25.5 | 99.9 | 99.9 | 47.0 | 100 | 95.4 | 100 | 100 |
| 1 | 27.7 | 99.7 | 99.9 | 50.4 | 99.9 | 90.7 | 100 | 100 |
| 2 | 57.6 | 99.9 | 100 | 76.2 | 98.1 | 94.9 | 99.9 | 100 |
| 3 | 86.5 | 100 | 99.9 | 78.9 | 93.2 | 98.4 | 99.9 | 100 |

(v) Instruction-incremental learning on the COCO Lin et al. (2014) dataset.

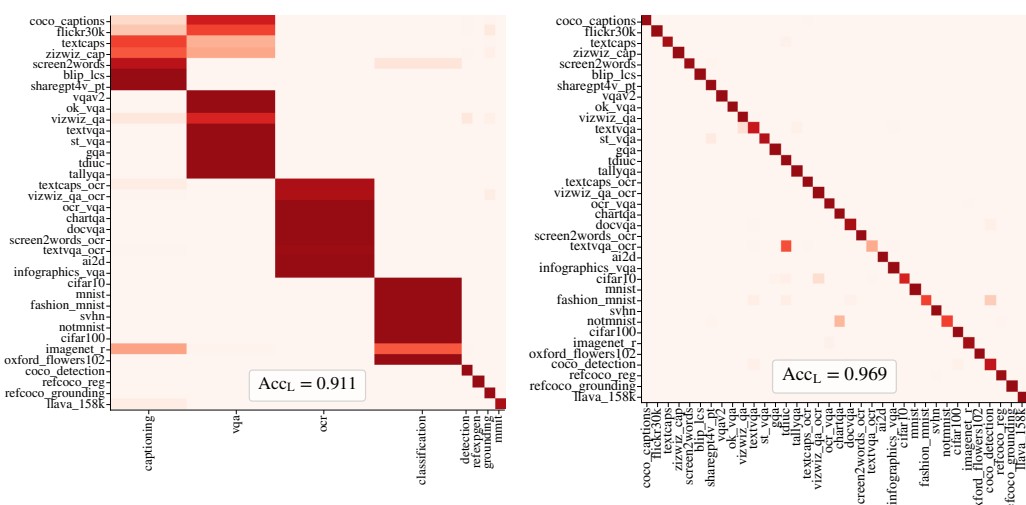

Figure 8: **Confusion matrix** for application-incremental learning across 8 applications (left) and 36 datasets (right).

## C ADDITIONAL ABLATION STUDY

**Task key representation and codebook depth for other incremental learning setups.** As shown in Tab. 17, we provide the lookup accuracy in other incremental learning setup. Using max-pooling consistently achieves desirable, if not optimal, selection accuracy.

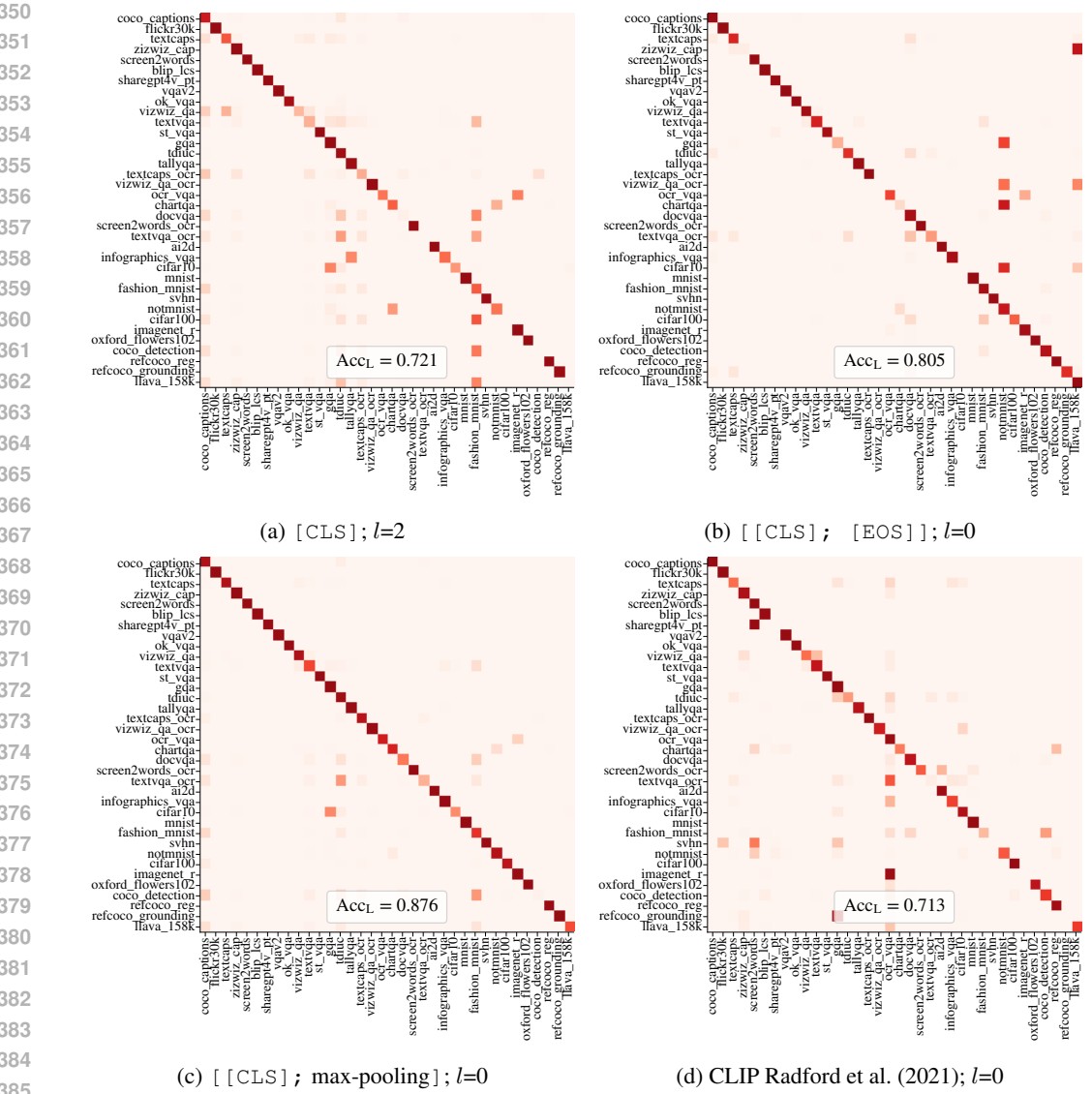

Figure 9: **Confusion matrix** for application-incremental learning across 8 applications (left) and 36 datasets (right).

**Confusion matrix.** In Fig. 8, we provide the confusion matrix for application-incremental learning experiments on 8 applications. The lookup accuracy is 91.1%, which is -5.8% lower than application-incremental learning on 36 datasets. Specifically, COCO Lin et al. (2014) and Flickr30K Young et al. (2014) datasets in the captioning application are mistakenly recognized as the VQA application, leading to undesirably low performances. This is potentially due to the shared task key representation for the captioning application being dominated by BLIP-LCS Li et al. (2022); Liu et al. (2023) and ShareGPT4V-PT Chen et al. (2023a) given the prevalence in example amount (1.4M *v.s.* 142K).

# D LIMITATIONS AND POTENTIAL NEGATIVE IMPACT

## D.1 LIMITATIONS

**Requirement for Task ID at Training Time.** One limitation of the proposed method is the requirement of the training task ID for task codebook adaptation. The adaptation process relies on identifying and retrieving the correct set of parameters at training time for a given task. This requirement poses a significant challenge in scenarios where the task ID may not be readily available or easily determined. It leaves for future work to explore training without the need of the task ID.

**Memory Overheads.** Although we aim to mitigate catastrophic forgetting without significant memory overheads, the scalability of the task codebook and its efficiency in a scenario with an extremely large number of tasks need further exploration. There could be practical limitations related to the memory footprint.

**Generalization across Diverse Domains.** While we evaluate the method across a broad range of tasks, the real-world applicability of the model's adaptability and efficiency in continuously evolving or extremely diverse domains remains to be examined. There might be domains or specific types of tasks where the proposed method does not perform as effectively.

## D.2 POTENTIAL NEGATIVE IMPACT

**Biases.** There is a risk of the model inheriting or amplifying biases present in the training data. The continuous adaptation process might not adequately address these biases, potentially perpetuating them in downstream applications.

**Privacy.** The adaptation of models to specific tasks could involve processing and storing sensitive or personal information, especially in applications that deal with user-generated content. If not properly handled, this could lead to privacy breaches and violations of data protection regulations.

**Environmental Impact.** Training and adapting large models require significant computational resources, which can have a considerable environmental footprint. Incremental adaptation could exacerbate this issue by requiring frequent retraining on new tasks.

