# OpenReview forum: "Incrementally Adapting Generative Vision-Language Models with Task Codebook"
_ICLR.cc/2025/Conference — Submitted to ICLR 2025_

### Official Review · Reviewer_SjeC · 2024-11-02

**Soundness:** 3
**Presentation:** 3
**Contribution:** 2
**Rating:** 3
**Confidence:** 4

**Summary:**

This paper focus on the continuously adapting vision-language models (VLMs). They assume that the tasks is sequential arrived. To solve this problem, this paper introduce a task codebook mechanism which contains task key and task value. Task key is used to identify the task and task value contains the parameters of adapter which is integrated with the base model for improving performance.

**Strengths:**

1. This is easy to follow. The readers can easily understand what problem this paper is trying to solve and how they do it.

2. This diagram is easy to understand.

3. This work also introduces a benchmark alongside the method, which completes this work.

**Weaknesses:**

1. I have a few concerns about this incremental learning setting: 1) what is the essential challenge of sequential? Current VLM will collect a vast amount of data covering a wide range of tasks. 2) With several new tasks, why don’t we finetune the model with all these tasks? This method shows better performance than TCIA. 3) I don’t quite understand what limits the model to performing multi-task fine-tuning. I still think that multi-task fine-tuning is a more feasible solution.

2. Is the task key necessary? Because the question is in text format, it’s easy to classify the task with the input question text. A lot of llm-based models can detect the task type with text.

3. In terms of novelty, the novelty of this work is limited. The tasks codebook is a collection of adapters, and the task key seems can be replaced by the inherent reasoning ability of lllms. The PEFT is normal in the finetuning community.

**Questions:**

See weaknesses

---

### Official Review · Reviewer_gpJh · 2024-11-02

**Soundness:** 4
**Presentation:** 4
**Contribution:** 3
**Rating:** 6
**Confidence:** 5

**Summary:**

This paper enables VLM models to handle multiple visual tasks through task codes, equipping the model with instruction-following capabilities while avoiding catastrophic forgetting in incremental learning. Additionally, the method provides a rich dataset for incremental learning.

**Strengths:**

1. This paper introduces a codebook to encode tasks, enabling the handling of different tasks during the inference phase and preventing catastrophic forgetting in incremental learning. I agree this approach is reasonable and interesting.

2. The article proposes a new multi-task dataset that covers various cross-modal tasks and different incremental settings.

3. The paper conducts extensive experiments to demonstrate the effectiveness of the method.

**Weaknesses:**

In my opinion, this paper's weakness lies in its overlap with the field of multimodal large language models (MLLMs). The authors need to clarify the differences from MLLMs, such as LLava, which excel not only in instruction-following but also show strong zero-shot performance in tasks like captioning, VQA, and OCR. Additionally, models like Ferret demonstrate localization capabilities.

To address this limitation more thoroughly, I suggest the authors:

Include a dedicated section comparing their approach to recent MLLMs like LLava and Ferret, highlighting key differences in architecture, training approach, and capabilities.
Conduct experiments to compare their method’s performance to these MLLMs on the proposed benchmark, especially for tasks where MLLMs have demonstrated strong zero-shot performance.
Discuss the potential advantages of their approach over MLLMs in incremental learning scenarios, if applicable.

**Questions:**

Please see the weakness.

---

### Official Review · Reviewer_9brX · 2024-11-07

**Soundness:** 3
**Presentation:** 3
**Contribution:** 2
**Rating:** 5
**Confidence:** 3

**Summary:**

In this paper, the authors introduced task codebooks to help VLMs adapt to multiple different and diverse tasks while minimizing forgetting. Their codebook involves using multiple task-specific MLPs (that act as values) that each corresponds to each task-specific key. They use the outputs of a certain layer predetermined by a hyperparameter $l$ as the guidance to learn the key that represents the tasks. Then, to determine the task during inference, they simply do nearest neighbor lookup with outputs from the same layer to the different MLPs. They also introduced a benchmark targeted to expose the catastrophic forgetting phenomenon amongst VLMs. The benchmark consists of 36 different datasets across 8 applications and 4 incremental learning setups. In their experiments, they are able to show how their method can successfully learn the correct tasks keys for almost all 36 different tasks. They also showed how their method outperforms models that are trained on multitasking and other anti-forgetting training methods such as prompt/prefix tuning.

**Strengths:**

The authors introduced an intuitive method to map task keys (outputs of a certain layer) to task values (the corresponding MLPs to deal with the tasks). Their benchmark also encompasses a variety of datasets and setups that surely exposes many models' incapabilities as they forget their knowledge when training workloads come sequentially. Their method also outperforms all other reported models that aim to tackle the same/similar issues, even when it comes to the popular prompt/prefix tuning methods.

**Weaknesses:**

1. The idea of using multiple different MLPs for each task sounds familiar when compared to the concept of Mixture-of-Experts (MoE). The paper did not mention this concept, nor did they compare.
2. The models evaluated are pretrained on close-source datasets including WebLI-100M, which is supposed to be a carefully curated high-quality dataset. It is possible that due to the difference in data quality, other models may underperform compared to the authors'.
3. Using nearest-neighbor lookup sounds simple, but when the number of tasks increases, so does the extra time that lookup incur during inference. In the paper, however, only lookup accuracy is discussed.
4. In the end, the paper focuses on "sequential" workloads. This demands that during training, every set of tasks come sequentially instead of randomly. In the experiments, however, the authors demonstrated themselves how using multitask models (pretrained where all training data across different tasks are randomized) mostly outperform their methods.

**Questions:**

1. The citation format is incorrect and makes the paper a bit hard to read. All the cited authors and years should have a pair of parentheses around them. Maybe you used \cite instead of \citep?
2. Typos: line 148: the two "processing" should be "processor".
3. There could be some sort of comparison to demonstrate how this method is better than or unique from using MoE to improve performance.
4. Is it possible that the models would perform better because of better training data? Is there a specific reason why you chose to use WebLI as the pretraining dataset?
5. It may be better to include a brief study on how much more time this process takes during training/inference.
6. This may be a more general question, but why would one prefer sequential workload over multitask/random workload during training?

---

### Meta-Review · Area_Chair_MuuW · 2024-12-19

**Metareview:**

This paper proposes to incrementally adapt generative vision-language models using a task codebook.  The paper received scores of 5,6,3.  The reviewers found some aspects of the approach interesting.  However, the critical issue that was raised was limited novelty.  However, there was no rebuttal.  The AC agrees with the reviewers' concerns, and recommends reject.

**Additional Comments On Reviewer Discussion:**

There was no discussion because no rebuttal was provided.

---

### Decision · Program_Chairs · 2025-01-22

Reject